power and energy systems/materials science/energy

diatomite, form-stable phase change material, reduced graphene oxide

**Author for correspondence:**
Min Li
e-mail: limin.li@163.com

This article has been edited by the Royal Society of Chemistry, including the commissioning, peer review process and editorial aspects up to the point of acceptance.

# Fabrication and characterization of capric acid/reduced graphene oxide decorated diatomite composite phase change materials for solar energy storage

## Min Li and Boyuan Mu

Jiangsu Key Laboratory for Construction Materials, Southeast University, Nanjing 211189, People's Republic of China

(iD) ML, 0000-0002-0616-5215

In this paper, diatomite-based composite phase change materials (DI-based CPCMs) were fabricated by the vacuum impregnation of capric acid (CA) into reduced graphene oxide decorated diatomite (rGO-DI). In the DI-based CPCMs, DI was used as the supporting material, which was first purified by thermal treatment and alkali treatment, to improve the adsorption capacity of the PCM, rGO was used to decorate the DI to improve the thermal conductivity of CPCMs. The rGO-DI could retain CA at the weight fraction of 60% without leakage. The maximum melting and freezing enthalpy of CA/rGO-DI-2 reached 106.2 J g$^{-1}$ and 108.6 J g$^{-1}$, respectively, and its thermal conductivity was up to 0.5226 W m$^{-1}$·K, 260.4% and 81.3% higher than pure CA and CA/DI, respectively. The CPCMs have good thermal reliability and thermal stability, and there was no chemical reaction between CA and rGO-DI. The CPCMs maintained thermal properties after 200 melting–freezing cycles. Finally, the CPCMs have potential for application in solar energy storage systems.

## 1. Introduction

Under the rapid depletion of unrecoverable energy sources and the continual enlargement of the gap between energy supply and demand, as well as environmental crisis, the use of various renewable energy sources has aroused huge attention [1,2]. Solar energy is a vital renewable energy source with the advantages of

being inexhaustible, cost-free and clean. It can be converted into clean energy by a series of conversions, especially by the photothermal conversion which is an efficient and direct application of solar radiation [3]. However, solar energy is a highly intermittent and unstable energy source. It is necessary to find a reliable and stable energy storage material to overcome the time and space difference of solar irradiation. Phase change materials (PCMs) are characterized by high-density heat storage [4]. Such materials can store and release thermal energy with much less temperature variation, having shown great potential application in thermal energy management and solar energy storage fields. PCMs can be any materials that are able to store or release energy from the melting or crystallization process. PCMs can be generally stratified into inorganic PCMs and organic PCMs. In comparison with inorganic PCMs, organic PCMs, such as paraffin and fatty acids, have aroused increasing attention because the organic PCMs are characterized by low price, non-toxicity and chemical stability. Moreover, the structures of organic PCMs do not vary during the repeated phase transition.

Fatty acid is a commonly used type of organic PCM. Fatty acid is characterized by high heat capacity, thermal stability, low supercooling and non-corrosiveness [5]. The commonly used fatty acids include capric acid (CA), lauric acid, palmitic acid, binary eutectic and ternary eutectic. The application of fatty acid has been extended in many fields, e.g. solar energy storage [6], greenhouse design [7], textiles [8] and air-conditioning systems [9]. Recently, the application of fatty acids for passive solar energy use in buildings has been widely investigated. However, fatty acids have certain undesirable properties, e.g. the leakage during the phase change and the extremely low thermal conductivity have limited the uses of passive solar energy use.

To prevent the leakage of PCM during the phase change, form-stable phase change materials (FSPCMs) were prepared by blending PCMs with supporting materials. Clay mineral-based materials with porous structure and high specific surface area broadly serve as supporting materials to prepared FSPCMs. Additionally, clay mineral-based FSPCMs have good compatibility with building materials, which means that clay mineral-based FSPCMs are promising materials in passive solar heating and building energy conservation. The commonly used clay mineral-based materials include expanded perlite [10], diatomite [11], kaolin [12], expanded vermiculite [13], etc. PCM is usually adsorbed into the interlayer or pores of the clay mineral-based materials. During the preparation, PCM has no chemical reaction with clay mineral-based materials. The major interactions between PCM and clay mineral-based materials include capillary force, surface tension and Van der Waal force. Diatomite (DI) serves as the supporting material owing to its light weight, highly porous structure, rigidity and relatively low price. Furthermore, the universality and chemical inertness of DI makes DI-based FSPCMs suitable for practical application [14–16]. Xu et al. [17] fabricated paraffin/DI composite phase change material (CPCM) at mix proportion (paraffin: DI) of 0.9 : 1.0. The DSC showed that the CPCMs had an excellent melting temperature of 41.11°C, and the latent heat capacity reached 70.51 J g$^{-1}$ under melting conditions. By incorporating polyethylene glycol (PEG) in the pores of diatomite, Karaman et al. [18]. prepared CPCM. In the CPCM, the highest content of PEG was 50 wt%, the melting and crystallization temperatures of the CPCM were 27.70°C and 32.19°C, respectively, and its latent heats were 87.09 J g$^{-1}$ and 82.22 J g$^{-1}$, respectively. However, the low adsorption capacity of DI and low thermal conductivity of DI-based FSPCM had effects on the thermal energy storage ability of FSPCMs for solar energy storage. The above problems should be addressed. To increase the content of PCM in the DI-based FSPCMs, several reports focused on improving the surface area and enlarging the pore of DI. Sun et al. [19] purified raw DI by performing thermal treatment. The adsorption ratio of composite paraffin in calcined DI was 61%, and the latent heat was 89.54 J g$^{-1}$. To produce the larger surface area without structural degradation, Lorwanishpaisarn et al. [20] treated DI with ultrasound. According to the result, the surface area of DI increased with the increase in treatment time to 60 min. Besides, the optimum ratio of paraffin in the treated DI was 60%.

Dispersion of micro/nanoparticles, e.g. carbonaceous materials, metals and metal oxides, has effectively increased the thermal conductivity of DI-based FSPCMs. Compared with metal and metal oxides, carbonaceous materials, e.g. expanded graphite [21], carbon nanotubes [22] and graphene [23], are characterized by high thermal conductivity, low density and high stability. In recent years, several research studies focused on using carbonaceous materials to improve the thermal conductivity of DI-based FSPCM. Qian et al. [24] studied the thermal properties of PEG/expanded graphite (EG)/ DI CPCM. According to the results, the maximum weight content of PEG was 55%, and the thermal conductivity of the CPCMs could increase from 0.36 W m$^{-1}$ · K to 0.71 W m$^{-1}$ · K after the addition of 10 wt% EG. Xu et al. [25] prepared paraffin/DI/multiwall carbon nanotube (MWCNT) CPCM. According to the results, the CPCM had a melting temperature of 27.12°C and phase change enthalpy of 89.4 J g$^{-1}$. Besides, the thermal conductivity of paraffin/DI/MWCNT CPCM significantly increased (42.45%) compared with that of paraffin/diatomite CPCM.

In the above research studies, the carbonaceous materials were added into DI-based CPCMs by mixing with PCM and DI. Thus, it was hard to achieve effective dispersion in CPCM as the carbon-based materials were easy to agglomerate due to high surface energy. Carbon-based materials decorate DI to effectively form a uniform layer of carbon on the surface of DI, increasing the thermal conductivity of DI and DI-based CPCM. Furthermore, the interaction between carbonaceous materials and PCM can reduce the negative effects of DI on the phase change enthalpy of CPCM [26,27]. To our best knowledge, no research has focused on improving the thermal conductivity by decorating DI to increase the thermal properties of DI-based CPCM. Graphene oxide (GO), a layered carbonaceous material, has become the focus of numerous investigations by its exceptional thermal properties and low cost [28], while its thermal conductivity is limited by the structural defects. To overcome this problem and deposit them on the surface of DI, the method to reduce the aqueous dispersion of GO to reduced graphene oxide (rGO) was used, which gained higher thermal conductivity under the reparation of structural defects. In this paper, the rGO was used to decorate diatomite. The reduced graphene oxide-diatomite (rGO-DI) was prepared by one-step sodium hydroxide solution treated with the GO and DI hybrid. The pores of DI were enlarged by alkali solution, increasing the adsorption amount to PCM. GO was reduced by alkali solution, and the rGO was decorated on DI surface, effectively increasing the thermal conductivity of DI and DI-based CPCM. Subsequently, a new type of DI-based CPCM was prepared by blending CA with rGO-DI. In this paper, the microstructure, chemical structure, thermal performance, thermal conductivity, thermal stability and reliability of CA/rGO-DI CPCMs were systematically investigated.

# 2. Material and methods

## 2.1. Materials

Sodium hydroxide (NaOH) (AR), hydrogen peroxide ($H_2O_2$) (30% w/w AR), sodium nitrate ($NaNO_3$) (AR), potassium permanganate ($KMnO_4$) (AR) and sulfuric acid ($H_2SO_4$) were purchased from Fisher Scientific. The diatomite sample was supplied from Pharmaceutical Industry Co., Ltd. Table 1 shows the chemical compositions of diatomite. Graphite powder was purchased from Qingdao Jinrilai Graphit Co. CA ($C_{10}H_{20}O_2$) was obtained from Shanghai Macklin Biochemical Co., Ltd.

## 2.2. Preparation of CA/rGO-DI CPCMs

GO was prepared from oxidation of graphite following a simplified Hummers method [29]. Crude DI was heat-treated for 2 h at 600°C to remove the organic impurities. The rGO-DI supporting material was prepared using a one-step method, DI and different mass of GO was added into 150 ml deionized water and ultrasonic treatment for 30 min. Subsequently, 3 g NaOH was added into the hybrid solution and magnetic stirred solution for 8 h at 80°C using a heating magnetic whisk to enlarge the pores of diatomite and react GO. Finally, the solution was filtered, washed with deionized water and dried to form rGO-DI supporting material. The rGO-DI supporting material was named rGO-DI-x, where × denotes the number of the sample. The formulation of the samples is listed in table 2. In the meantime, 100 mg GO was added into 150 ml deionized water, ultrasonic treatment was performed for 30 min. Also, 3 g NaOH was added into GO solution and magnetic stirred hybrid solution for 8 h at 80°C to prepare rGO.

The CA/DI and CA/rGO-DI-x CPCMs were prepared using the vacuum impregnation method. A certain amount of solid CA was added into the DI or rGO-DI-x, put into a vacuum drying oven and heated to 70°C for 1 h. The mass content of CA in the CPCMs was 55 wt%, 60 wt% and 65 wt%, respectively. The preparation process is shown in figure 1.

## 2.3. Characterization

The surface morphology of DI and CA/rGO-DI CPCM was obtained by a scanning electron microscope (SEM, ZEISS Company, Germany) at 5 kV. The elemental analysis was accomplished by an energy-dispersive spectroscope (EDS) attached to an SEM. Fourier transform spectroscopy (FT-IR) spectra groups of the pure CA and CA/rGO-ID CPCM were recorded in the transmission mode on an FTIR spectrometer (IS10, Nicole Company, USA) at the wavenumber range of 4000–400 cm$^{-1}$. The crystalline structures of CA and CA/rGO-DI CPCM were characterized by X-ray diffraction

**Table 1.** Chemical compositions (wt%) of diatomite used in this study.

| constituent | $SiO_2$ | $Al_2O_3$ | $Fe_2O_3$ | MgO | CaO | $Na_2O$ | $K_2O$ | $TiO_2$ | other |
|---|---|---|---|---|---|---|---|---|---|
| ratio (%) | 80.21 | 10.26 | 2.18 | 0.88 | 0.94 | 2.22 | 0.85 | 0.36 | 2.1 |

**Table 2.** The formulation of rGO-DI-x CPCMs.

| | rGO-DI-1 | rGO-DI-2 | rGO-DI-3 |
|---|---|---|---|
| mass of DI (g) | 10 | 10 | 10 |
| mass of GO (mg) | 100 | 150 | 200 |

measurements (Smart Lab 3, RIGAKU, Japan). The melting temperature, freezing temperatures as well as phase change enthalpy of CA and CA/rGO-DI CPCM were analysed by a differential scanning calorimeter (DSC 204 F1, Germany) at $5°C\ min^{-1}$ under nitrogen atmosphere. The weight loss and thermal stability of samples were investigated by a thermal gravimetric analyser (Netzsch STA449 F3, Germany) with the range of room temperature to 600°C at a heating rate of $10°C\ min^{-1}$ under a nitrogen atmosphere. The thermal conductivities of the CA, rGO/DI and CA/rGO-DI-x CPCMs at 25°C were tested by a thermal constant analyser (TPS2500S, Hot Disk, Sweden).

# 3. Results and discussion

## 3.1. Characterization and structure of rGO

FT-IR, Raman and TG analysis were used to characterize the chemical structure of rGO. The FT-IR spectrum was recorded to characterize the structure of the GO and rGO. In the spectra of GO, the stretching vibration absorption peak in the range of $3200–3550\ cm^{-1}$ is the –OH on GO. At $1768\ cm^{-1}$, there is a C=O stretching vibration absorption peak in the –COOH group, and $1626\ cm^{-1}$ is a C=C vibration absorption peak. $1026–1300\ cm^{-1}$ is the stretching vibration peak of C–O and C–O–C (C–OH) on the epoxy group, and $1379\ cm^{-1}$ is the deformation peak of -OH [30]. In the spectra of the rGO, the intensity of the peaks at $1768\ cm^{-1}$, $1626\ cm^{-1}$ and $1026–1300\ cm^{-1}$ was weakened or disappeared, which indicated that GO was chemically reduced.

Figure 2b shows the Raman spectra of the GO and rGO. All these exhibit D ($1341.1\ cm^{-1}$) and G ($1590.7\ cm^{-1}$) bands, which are associated with sp2 carbon atoms in the hexagonal carbon framework and sp3-hybridized carbon atoms at the edges or defects on the graphene basal plane, respectively. The intensity ratio of D band and G band ($I_D/I_G$) can be estimated by the number of defects and disorder in carbon materials. Higher $I_D/I_G$ values for the rGO compared to the GO can be attributed to the formation of more sp3 and sp2 graphitic lattice after reduction.

The TG analyses of the GO and rGO are shown in figure 2c. Significant mass loss occurred for the GO between 170°C and 250°C, which was related to the pyrolysis of the labile oxygen-containing functional groups during the heating process. The residual quantity of GO was 42.5%, indicating that the content of graphene was approximately 40%. In addition, the residual quantity of rGO was 87.2%. The residual quantities of rGO confirmed that most of oxygen-containing groups of GO had been reduced and there were some oxygen-containing group residues in the rGO sample. Based on the results from the FT-IR, Raman and TG analyses, we conclude that rGO had been prepared being stirred in NaOH solution for 8 h at 80°C.

## 3.2. Shape-Stable properties of CA/DI and CA/rGO-DI CPCM

The macroscopic photographs of CA/DI and CA/rGO-DI-2 CPCM with different contents of CA are shown in figure 3a; it is obvious that the composite PCMs were agglomerate when the content of CA was 65 wt%. To investigate the exudation stability, CA/DI and CA/rGO-DI-1 CPCM were heated to 80°C for 0–2 h, the residual quantity of CA/DI and CA/rGO-DI-1 CPCM is shown in figure 3b. The mass loss during the heating process can be neglected when the content of CA was 55 wt% and 60 wt%.

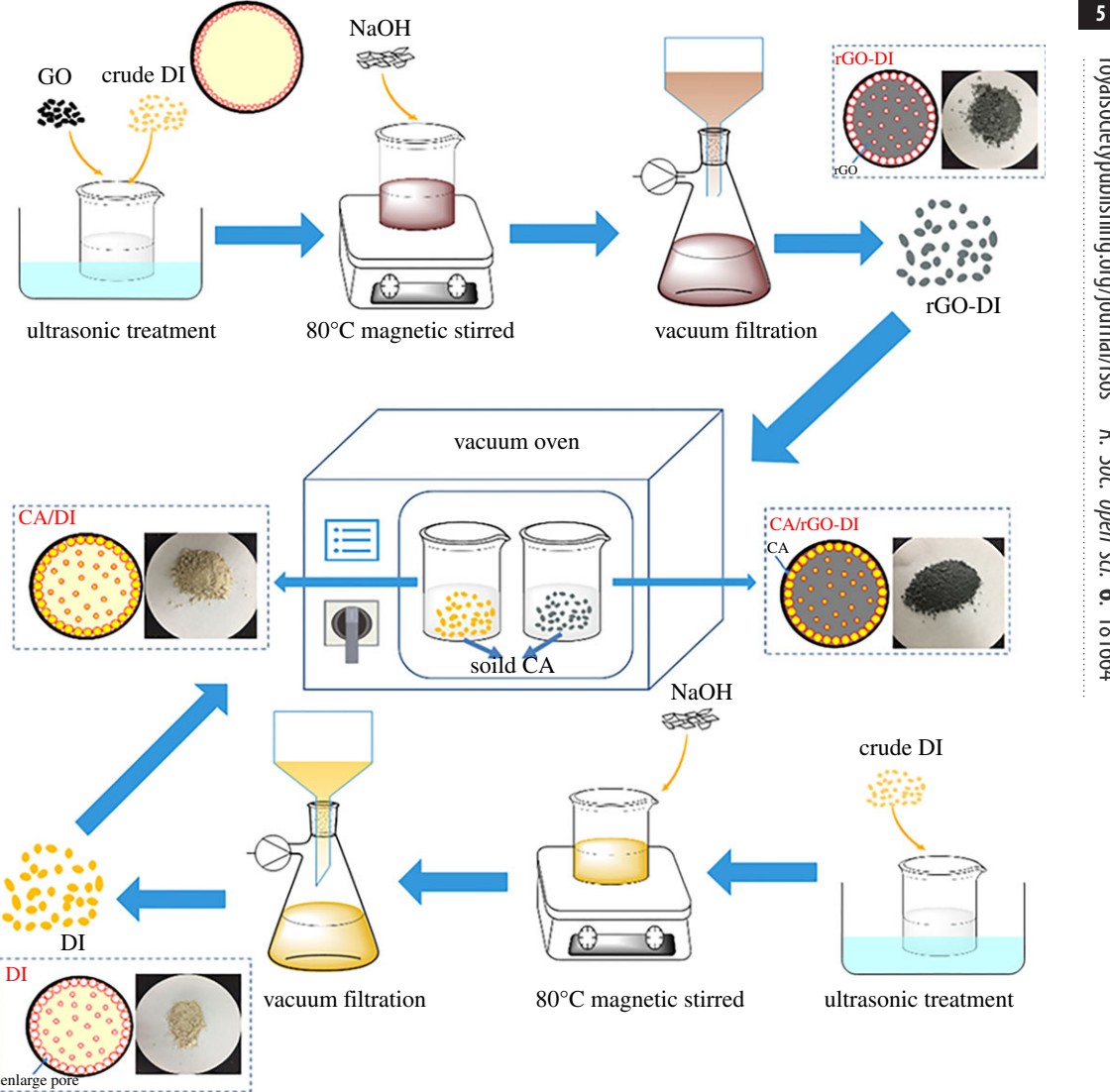

**Figure 1.** Preparation process of CA/DI and CA/rGO-DI CPCM.

Yet, significant mass loss (about 10 wt%) occurred for CA/DI and CA/rGO-DI-1 CPCM when the content of CA was 65 wt%. Thus, the optimum mass fraction of CA in composite without leakage is 60 wt%.

## 3.3. Morphologies and microstructures of DI, rGO-DI and CA/rGO-DI CPCMs

The microstructures of DI, rGO-DI and CA/rGO-DI CPCM were investigated using SEM and EDS analyses. The relevant results are shown in figures 4 and 5. Figure 4a,b shows the morphology structures of DI. It is shown that DI is primarily composed of disc-like structures as well as several pores at the surface of DI. The modified DI with alkali treatment is shown in figure 4c,d. A considerable number of pores at the surface of rGO-DI were enlarged because the impurities were removed, and alkali solution reacted with the $SiO_2$, the main material of the inner walls of pores. As a result, the adsorption capacity of rGO-DI could be improved, as shown in figure 5, the EDS analysis was clear that the $SiO_2$ content of DI significantly decreased after alkali treatment. Figure 4e,f shows that CA completely filled the available pore volume in rGO-DI, and CA was closely integrated with the DI. Because of the excellent wetting property of rGO with CA, the interfaces of CA and the rGO-DI were combined tightly. In addition, the capillary force between PCM and supporting materials could avoid the leakage of CA when phase change occurred.

The XRD patterns of CA, DI, rGO-DI, CA/DI and CA/rGO-DI-2 CPCM are shown in figure 6. Figure 6a shows that CA shows the complete crystalline structure, and all the sharp and strong peaks correspond to a single phase of CA. Figure 6b shows that the broad peaks of DI and rGO-DI at nearly

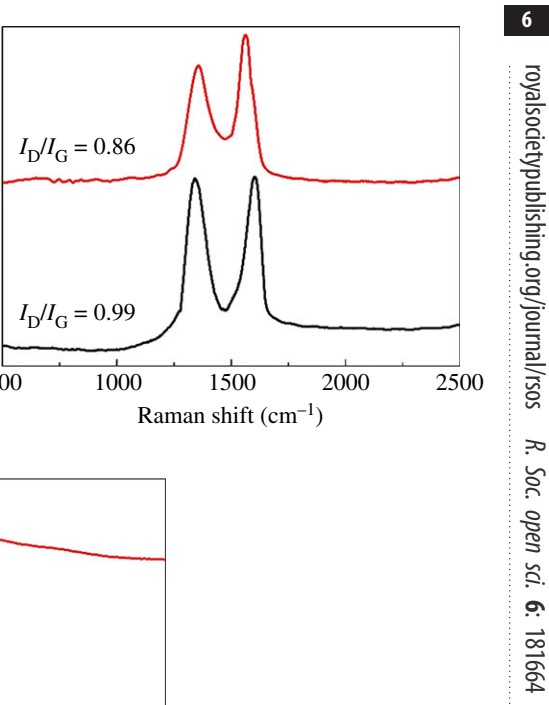

**Figure 2.** (a) FT-IR spectra, (b) Raman spectra and (c) TG analysis of GO and rGO.

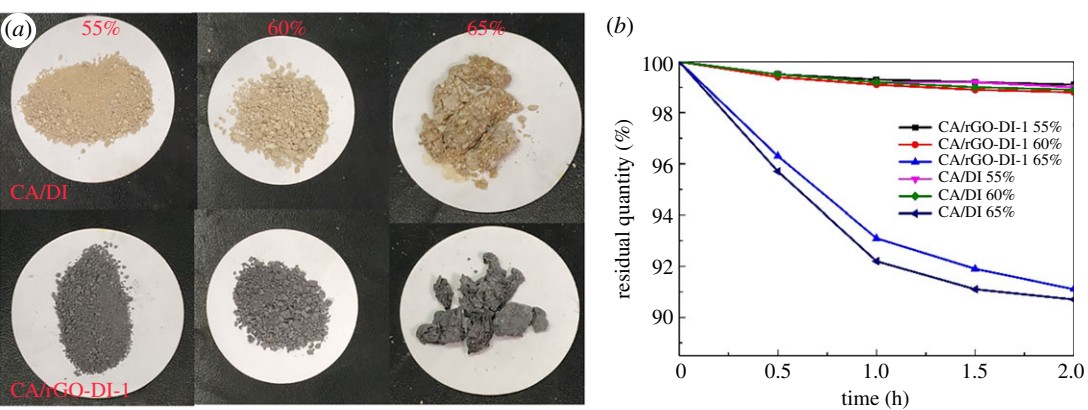

**Figure 3.** Photographs of CA/DI and CA/rGO-DI-1 CPCM (a); The residual quantity of the composite PCMs during the heating process (b).

$22°$ show a typical non-crystalline structure of diatomite. The XRD curves of DI and rGO-DI are similar, suggesting the crystalline structure of DI was not changed by rGO. The characteristic peaks of the rGO-DI and CA in the CA/rGO-DI CPCM are both shown in the XRD curves of CA/DI and CA/rGO-DI CPCM. The sharp diffraction peaks of CA/DI and CA/rGO-DI CPCM around $9°$, $11°$, $19°$ and $22°$ are assigned to CA crystal, and the broad peak of rGO-DI is between $20°$ and $25°$. This indicates that the crystal structure of CA was not destroyed after the impregnation.

## 3.4. Chemical characterizations of CA and CPCM

To study the chemical compatibility of the CA/rGO-DI CPCMs, the FT-IR spectrum experiment was performed to characterize the structure of CA, DI, rGO-DI and CA/rGO-DI-2 CPCM. According to the FT-IR spectrum of CA in figure 6, the peaks at $2957 \, cm^{-1}$, $2909 \, cm^{-1}$ and $2851 \, cm^{-1}$ are the stretching

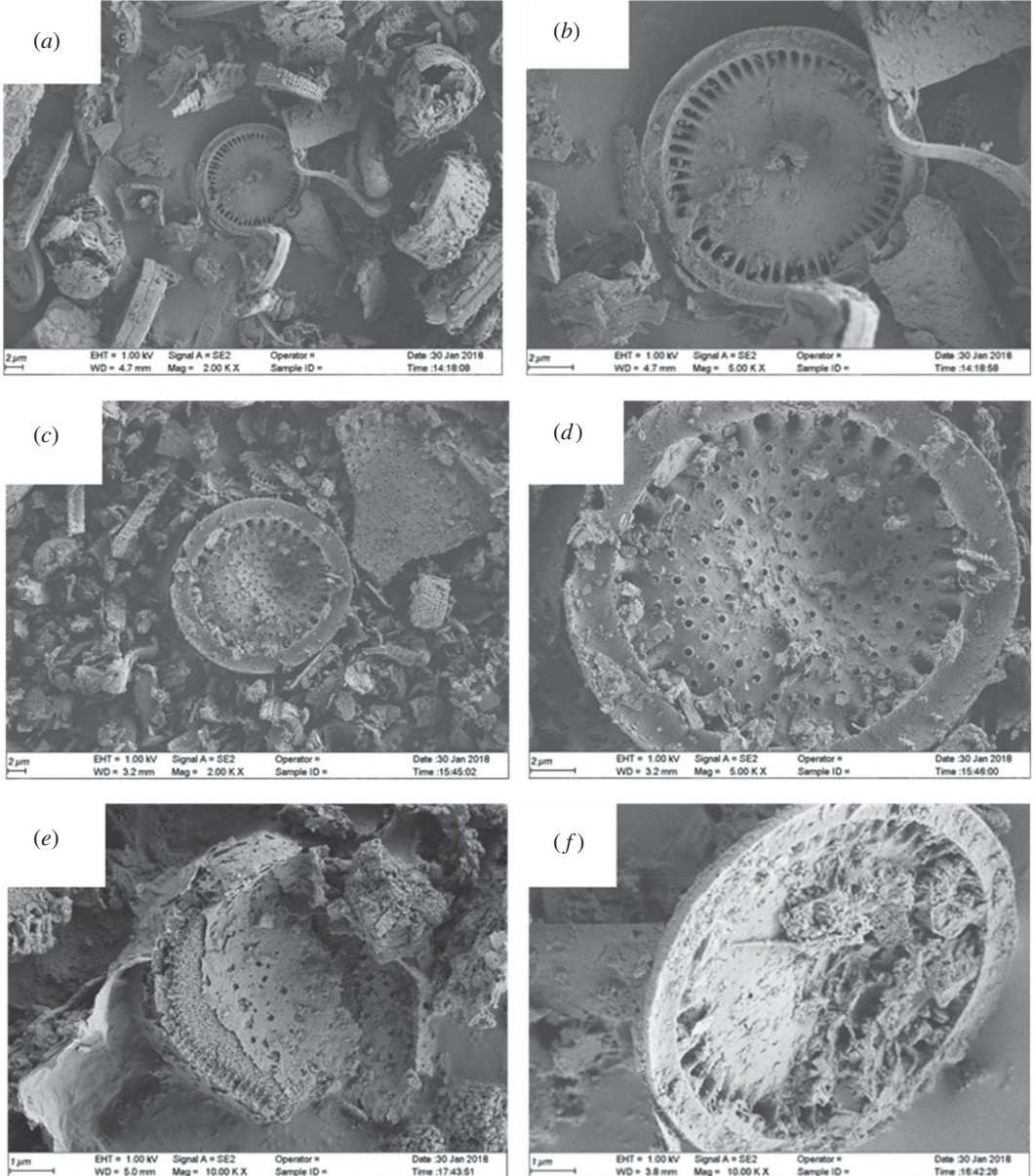

**Figure 4.** SEM images of DI (*a,b*), rGO-DI (*c,d*), CA/rGO-DI (*e,f*) CPCM.

vibrations of C–H coming from –CH$_3$ and -CH$_2$. The peak at 1714 cm$^{-1}$ is the stretching vibration of C = O located at the edge of CA. In the FT-IR spectrum of DI, the peaks at 801 cm$^{-1}$ and 1062 cm$^{-1}$ are the asymmetric stretching mode symmetric stretching vibration of the Si-O-Si group. The peak at 3437 cm$^{-1}$ is attributed to the Si-OH groups. All these peaks suggest that DI is primarily composed of SiO$_2$. Many absorption peaks in the FT-IR spectra of DI and rGO-DI are similar. There were only two differences between the FT-IR spectrum of DI and rGO-DI. The peaks at 2916 cm$^{-1}$ and 2867 cm$^{-1}$ reflect the C–H bond of rGO. The other is the peak of rGO-DI at 1625 cm$^{-1}$, which is the C–OH group located at rGO. The characteristic peaks of both CA and rGO-DI are observed in the FT-IR spectrum of CA/rGO-DI-2. Figure 7 shows that rGO was located at the surface of DI, CA had just been physically absorbed into rGO-DI. Furthermore, no chemical interaction occurred between CA and rGO-DI.

## 3.5. Thermal conductive characteristics of CA, CA/DI and CA/rGO-DI CPCMs

Thermal conductivity is a key parameter, associated with the thermal energy storage and release rates of CPCMs directly. The thermal conductivities of DI, rGO-DI, CA/DI and CA/rGO-DI CPCMs are shown in figure 8. It is shown that the thermal conductivities of CA and CA/DI were 0.1450 W m$^{-1}$·K and 0.2882 W m$^{-1}$·K, respectively, while those for the CA/rGO-DI-1, CA/rGO-DI-2 and CA/rGO-DI-3 were

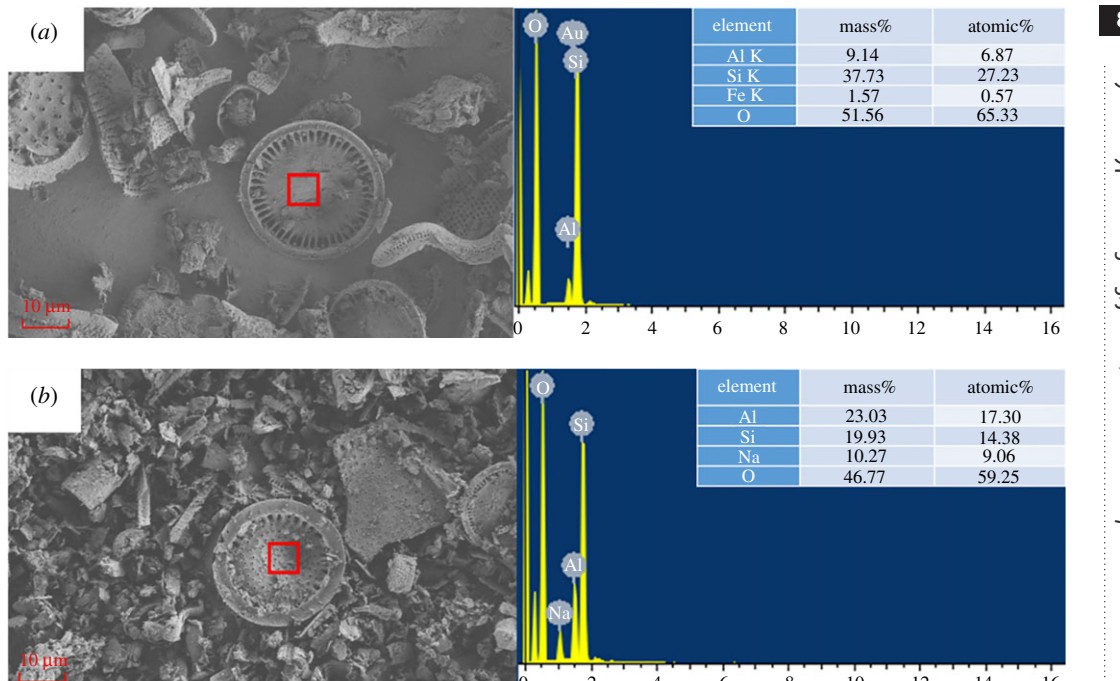

**Figure 5.** EDS analysis of the DI (*a*) and rGO-DI (*b*).

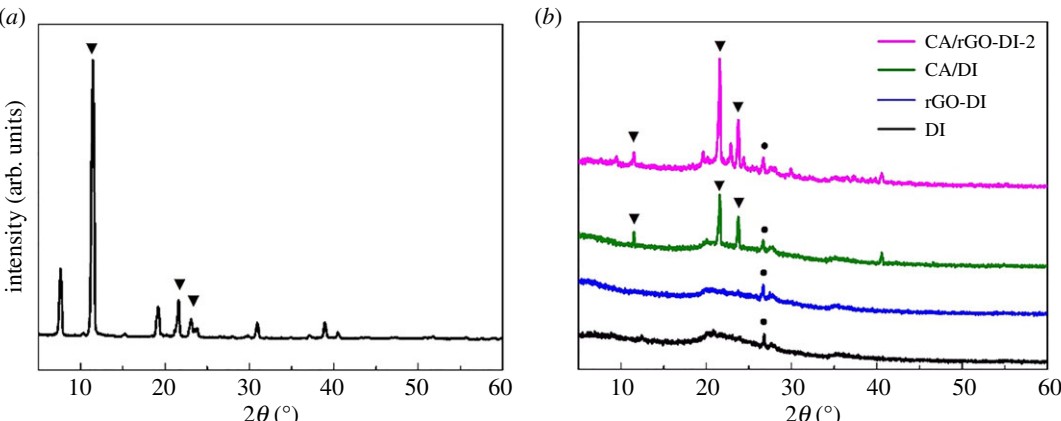

**Figure 6.** XRD patterns of CA (*a*), DI, rGO-DI, CA/DI and CA/rGO-DI-2 CPCM (*b*).

0.445 W m$^{-1}$ · K, 0.5226 W m$^{-1}$ · K and 0.5693 W m$^{-1}$ · K, respectively. The thermal conductivity of CA/DI and CA/rGO-DI-x was 98.8%, 206.9%, 260.4% and 292.6% higher than pure CA. Furthermore, these results correspond to the increase in the thermal conductivity of CA/DI CPCMs after decorating rGO around 54.4%, 81.3% and 97.5%, respectively. The enhancement of the thermal conductivity of CPCMs is through a two-step process. The enhancements were first observed when CA was impregnated in DI pores due to the relatively high thermal conductivity of DI. The enhancements were mainly ascribed to the decorated rGO and their effective dispersion on the surface of DI. It is well known that graphene-based materials have a particularly high thermal conductivity. The thermal conductivity of DI can be improved by rGO decorating. In addition, the thermal conductivity became higher with increasing content of rGO, further enhancing the thermal conductivity of CA/rGO-DI CPCMs. As the content of rGO in the CPCMs is less than 1 wt%, thermal conductivity is significantly improved.

## 3.6. Thermal properties of CA and CPCMs

Temperature and latent heat of phase change as well as supercooling degree are critical thermal parameters of the CA/rGO-DI CPCMs. The thermal properties of pure CA, CA/DI CPCM and CA/

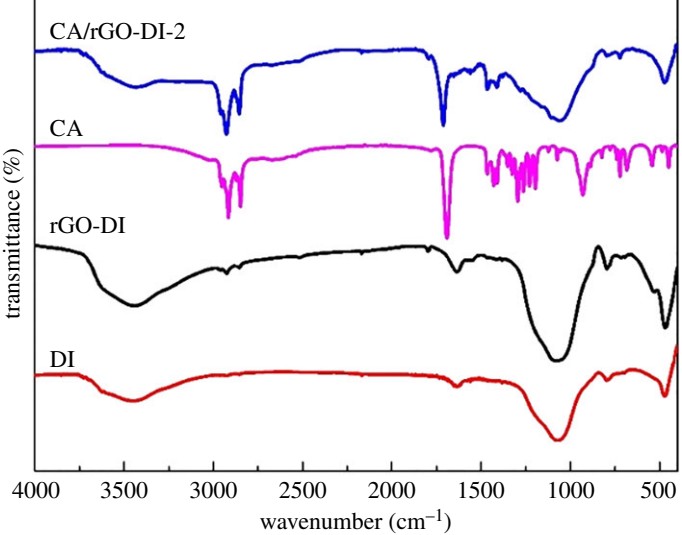

**Figure 7.** FT-IR spectra of DI, rGO-DI, CA and CA/rGO-DI-2 CPCM.

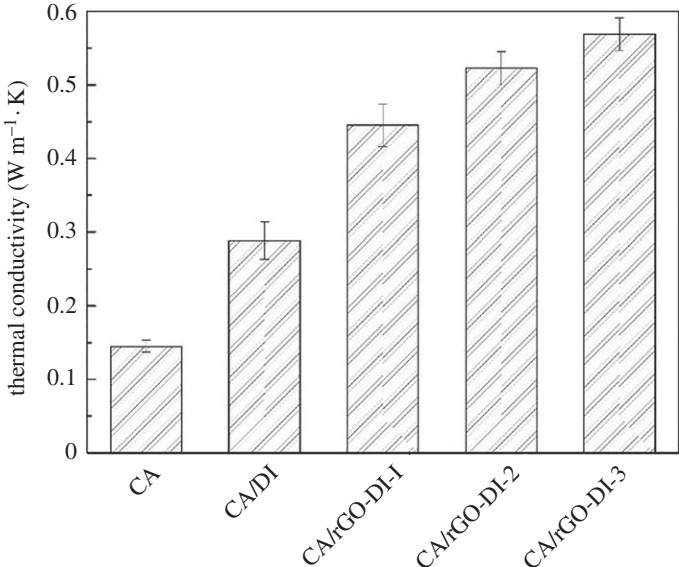

**Figure 8.** Thermal conductivity of the CA, CA/DI and CA/rGO-DI CPCM.

rGO-DI CPCMs were analysed by DSC, and the curves are shown in figures 9 and 10. The resulting DSC data are listed in table 3. In figure 9, the DSC results of pure CA suggested that the melting temperature ($T_m$) and the freezing temperature ($T_f$) of CA are 31.3°C and 28.3°C, respectively, and the corresponding phase change enthalpies are 187.7 J g$^{-1}$ and 190.1 J g$^{-1}$, respectively. Figure 10a,b shows that the DSC curves of both melting and freezing curves are similar to those of CA. With the addition of the rGO, the melting and freezing latent heat of the CA/rGO-DI CPCMs was higher than that of CA/DI CPCM. According to these results, the crystal structure of CA was not destroyed after impregnation. Furthermore, as the crystallinity of CA was improved in rGO-DI, $\Delta H_m$ and $\Delta H_f$ of CA/rGO-DI CPCMs were higher than those of CA/DI CPCM. The highest $\Delta H_m$ and $\Delta H_f$ are 106.2 J g$^{-1}$ and 108.6 J g$^{-1}$, respectively, located at CA/rGO-DI-2 CPCM. The theoretical latent heat of CPCM can be calculated by the following equation [31]:

$$\Delta H_T = \Delta H_{CA} * \eta \; (\text{wt.\%}), \tag{3.1}$$

where $\Delta H_T$ is the theoretical latent heat of CA/rGO-DI CPCMs, $\Delta H_{CA}$ is the latent heat of pure CA and $\eta$ is the mass fraction of CA within the CPCMs. Table 2 shows that the theoretical latent heat capacity of CA/rGO-DI CPCMs is higher than the measured value, and the phase change temperatures of all

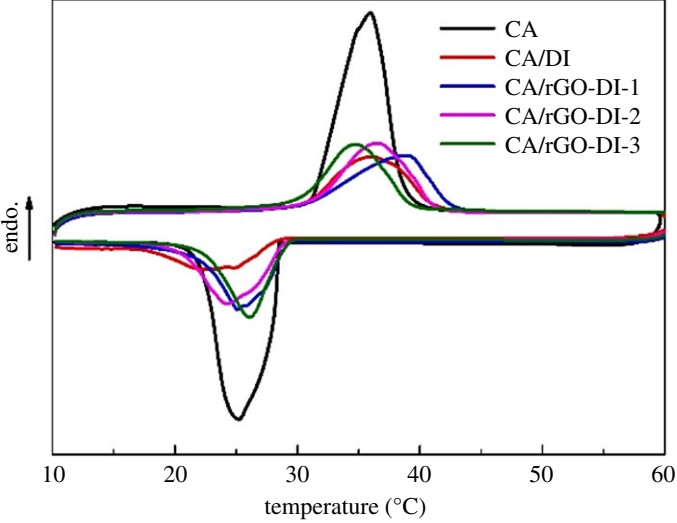

**Figure 9.** DSC curves of CA, CA/DI and CA/rGO-DI CPCM.

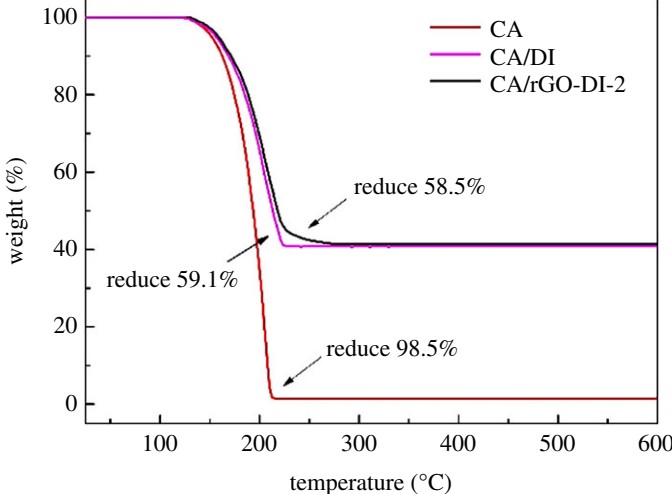

**Figure 10.** TG curves of CA CA/DI and CA/rGO-DI-2 CPCM.

CA/rGO-DI CPCMs and CA/DI CPCM were lower than those of pure CA. This is because the crystal arrangement and orientation of CA chains in the porous matrices of DI were restricted, resulting in a decrease in the regularity of the crystalline regions [32]. Furthermore, with the increase in rGO, the melting temperature of the CPCMs fell, and the freezing temperature of the CPCMs rose. The supercooling of CA/rGO-DI CPCMs was lower than pure CA and CA/DI CPCM, suggesting that the good thermal conductivity of the rGO-DI accelerated the melting and freezing behaviour of the CA.

Moreover, table 4 draws the comparison of the energy storage performance of the CA/rGO-DI CPCMs with that of some CPCMs reported in the literature, and the CA/rGO-DI CPCMs shows the obvious advantages over the previous reports [19,24,33–40]. This suggests that the CA/rGO-DI CPCMs have great potential application in thermal storage.

## 3.7. Thermal stability of CA and CA/rGO-DI CPCM

For thermal storage applications, it is highly necessary to test the thermal stability of CA/rGO-DI CPCM. The TG curves obtained for CA, CA/DI CPCM and CA/rGO-DI-2 CPCM are shown in figure 11. CA shows one-step degradation that started at around 125°C and ended at 210°C. According to the curves of CA/DI and CA/rGO-DI-2, the initial decomposition temperature was higher than that of CA, which indicates that the DI and rGO-DI improved the thermal stability of CA. The weight loss percentages of CA/DI and CA/rGO-DI-2 are 59.1% and 58.5%, respectively. Given the errors, the

**Table 3.** Thermal properties of CA, CA/DI and CA/rGO-DI CPCMs.

| samples | melting process | | freezing process | | |
|---|---|---|---|---|---|
| | $T_m$ (°C) | $\Delta$Hm (J g$^{-1}$) | $T_f$ (°C) | $\Delta$Hf (J g$^{-1}$) | supercooling (°C) |
| CA | 31.3 | 187.7 | 28.3 | 190.1 | 3.0 |
| CA/DI | 30.9 | 94.5 | 28.6 | 89.1 | 2.3 |
| CA/rGO-DI-1 | 30.6 | 103.9 | 29.2 | 104.2 | 1.4 |
| CA/rGO-DI-2 | 30.5 | 106.2 | 29.2 | 108.6 | 1.3 |
| CA/rGO-DI-3 | 30.0 | 98.6 | 29.5 | 98.4 | 0.5 |

**Table 4.** Comparison of DI-based CPCMs in the literature.

| sample | melting temperature (°C) | latent heat of melting (J g$^{-1}$) | freezing temperature (°C) | latent heat of freezing (J g$^{-1}$) | reference |
|---|---|---|---|---|---|
| paraffin/calcined diatomite | 33.04 | 89.54 | — | — | [19] |
| polyethylene glycol/ diatomite | 59 | 103.7 | 40 | 92.08 | [24] |
| caprice – myristic acid/diatomite | 16.74 | 66.81 | — | — | [33] |
| lauric acid/diatomite | 40.7 | 43.2 | 37.6 | 42.38 | [34] |
| paraffin/layered kaolin | 50.57 | 94.88 | 53.65 | 93.0 | [35] |
| lauric acid/kaolin | 43.7 | 72.5 | 39.3 | 70.9 | [36] |
| paraffin/EP | 42.27 | 87.40 | 40.79 | 90.25 | [37] |
| CA-stearic acid/ EP | 29.6 | 82.1 | 17.4 | 82.6 | [38] |
| LA-PA-SA/ vermiculite | 31.4 | 75.8 | 30.3 | 73.2 | [39] |
| paraffin/vermiculite | 27.0 | 77.6 | 25.1 | 71.5 | [40] |
| CA/rGO-DI-2 | 30.5 | 106.2 | 29.2 | 108.6 | present study |

residual quantity is well consistent with the nominal additive amount of CA in the composites. The data imply that the CA and rGO-DI, CA and DI were mixed successfully. The earlier occurrence of thermal degradation of CA compared with CA/DI CPCM and CA/rGO-DI-2 CPCM suggests that the CPCMs had higher thermal stability. It can be found that the weight loss of the CA/DI CPCM and CA/rGO-DI-2 CPCM was less than 0.1% at the temperature below 100°C. This result suggests that the CPCMs had excellent thermal stability.

## 3.8. The thermal reliability analysis of CA/rGO-DI CPCM

The thermal reliability and chemical stability of the CPCMs through many melting-freezing cycles are critical factors to determine the suitability of the CPCM for energy storage applications. To investigate the thermal reliability and chemical stability of CPCM, 200 thermal cycles tests were performed. DSC curves of the CA/rGO-DI-2 CPCM after thermal cycles are shown in figure 12, and the detailed information is summarized in table 5. It is found that melting and freezing temperature decreased slightly, the melting and freezing temperatures of CA/rGO-DI-2 CPCM were 29.4°C and 27.7°C after the experiment, and those of CA/rGO-DI-2 CPCM were 30.5°C and 29.2°C before the thermal cycles. Moreover, the melting and solidifying latent heat of the CA/rGO-DI-2 were 100.5 J g$^{-1}$ and 101.5 J g$^{-1}$ after thermal cycles, corresponding to the slight decrease of 5.41% and 6.51%, respectively. The variation of latent heat and phase change temperature was the acceptable value for application.

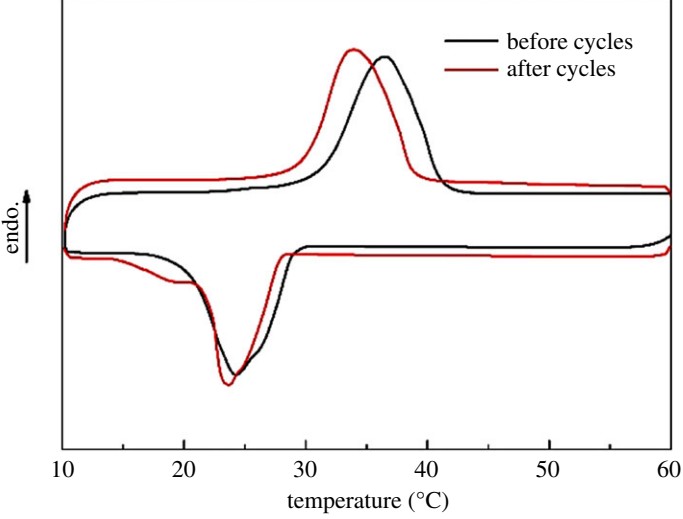

**Figure 11.** DSC curves of the CA/rGO-DI-2 CPCM before and after 200 thermal cycles.

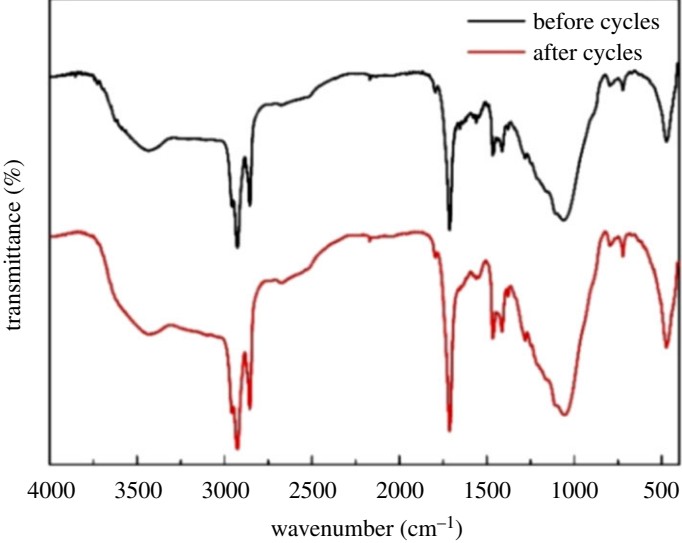

**Figure 12.** FT–IR spectrum of the CA/rGO-DI-2 CPCM before and after 200 thermal cycles.

Besides, the chemical stability of the CA/rGO-DI-2 CPCM after thermal cycles is shown in figure 13. The chemical stability was examined using FT-IR analysis, and the FT-IR curves of the CA/rGO-DI-2 CPCM after the thermal cycles are shown in figure 13. Even after 200 melting-freezing cycles, there was no detected variation in the location and shapes of IR spectrum peak of the CA/rGO-DI-2 CPCM. These results verify that the CA/rGO-DI-2 CPCM had excellent thermal reliability and chemical stability.

## 3.9. Heat storage and release experiments of CA/DI and CA/rGO-DI CPCMs

The temperature–time curves were employed to analyse the effects of thermal conductivity on the melting and freezing times. Figure 14*a* shows that the heat storage time initiated at 15°C and ended at 45°C. It took more than 3000 s to heat CA/DI CPCM from 15°C to 45°C, the temperature plateau is at around 26–29°C. The melting of CA/rGO-DI CPCMs lasted from 2200 s to 2500 s, and the temperature plateau appeared at 26–30°C. Figure 14*b* shows that when temperature decreased from 45°C to 15°C, it took about 5000 s for CA/DI and the temperature plateau appeared at 25–28°C, the freezing temperature of CA/DI CPCM. CA/rGO-DI CPCMs had the similar phase change temperature plateau, and the freezing of CA/rGO-DI lasted from 2500 s to 3500 s. The melting and freezing temperature plateaus of CA/DI and CA/rGO-DI CPCMs correspond to the phase change temperature from DSC analysis. In addition, the length of temperature plateau of CA/rGO-DI CPCMs

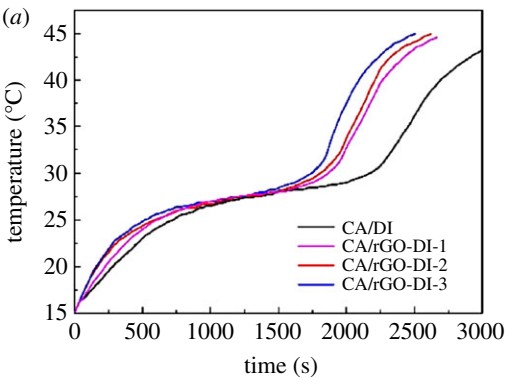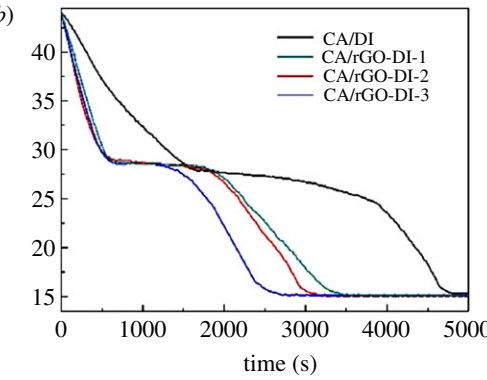

**Figure 13.** The melting curve (*a*) and freezing curve (*b*) of CA/DI and CA/rGO-DI CPCMs.

**Table 5.** Thermal properties of the CA/rGO-DI-2 CPCM before and after thermal cycles.

| | melting process | | freezing process | |
|---|---|---|---|---|
| sample | $T_m$ (°C) | $\Delta Hm$ (J g$^{-1}$) | $T_f$ (°C) | $\Delta Hf$ (J g$^{-1}$) |
| before cycles | 30.5 | 106.2 | 29.2 | 108.6 |
| after cycles | 29.4 | 100.5 | 27.7 | 101.5 |

correspond to the thermal conductivity of CA/rGO-DI CPCMs. When the thermal conductivity increased, the temperature plateau was shortened. The latent heat of phase change of CA/DI CPCM and CA/rGO-DI CPCMs was similar so that the greater thermal conductivity of CA/rGO-DI CPCMs was the primary reason why the phase change times of CA/rGO-DI CPCMs were shorter than those of CA.

# 4. Conclusion

A new type of CPCM was prepared with CA and rGO decorated diatomite by vacuum impregnation. The thermal properties of CA/rGO-DI CPCMs were studied in detail. The conclusions were as follows:

(1) The rGO was decorated on the surface of DI successfully, and the optimum mass fraction of CA and DI in the CPCMs was 60 : 40. The SEM result suggests that CA was well incorporated into rGO-DI pores. The XRD and FT-IR results suggest that there were only physical interactions between CA and rGO-DI. The TG results proved that the thermal stability of CA/rGO-DI CPCM was better than that of pure CA.

(2) The thermal conductivity of CA/rGO-DI PCM increased with the decorated amount of rGO. The thermal conductivity of CA/rGO-DI-3 CPCM reached 0.5693 W m$^{-1}$·K, which was enhanced by 292.6% and 97.5% compared to the CA and CA/DI CPCM, respectively. The reduced melting and freezing time indirectly confirmed the results.

(3) The $\Delta H_m$ and $\Delta H_f$ of CA/rGO-DI-2 were 106.2 J g$^{-1}$ and 108.6 J g$^{-1}$, respectively, and the CA/rGO-DI CPCMs showed lower supercooling than that of CA/DI CPCM. The $T_m$ and $T_f$ of CA/rGO-DI-2 CPCM were 30.5°C and 29.2°C, respectively. Furthermore, the latent heat and chemical structure had changed little even after a 200-cycle test.

Therefore, the prepared CA/rGO-DI CPCMs showed excellent thermal properties and had potential applications in solar energy storage systems.

Data accessibility. There are no additional data to accompany this manuscript. All data generated or analysed during this study are included in this manuscript.

Authors' contributions. L.M. conceived and designed the experiments, coordinated the study and helped draft the manuscript. B.M. performed the experiments and analysed the results. All the authors reviewed and approved the final manuscript.

Competing interests. The authors declare that they have no competing interests.

Funding. This work was supported by the National Natural Science Foundation of China (51178102) and the key project of National Natural Science Foundation of China (51738003), which is financed by the National Natural Science Fund Committee.

Acknowledgements. The authors thank the editors and anonymous reviewers for their helpful suggestions for this manuscript.

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
