## [Reviewer comments · Royal Society Open Science]

Review History

RSOS-181664.R0 (Original submission)

Review form: Reviewer 1 (Yibing Cai)

Is the manuscript scientifically sound in its present form?

Yes

Are the interpretations and conclusions justified by the results?

Yes

Is the language acceptable?

Yes

Is it clear how to access all supporting data?

Yes

Do you have any ethical concerns with this paper?

No

Have you any concerns about statistical analyses in this paper?

No

Recommendation?

Accept with minor revision (please list in comments)

Comments to the Author(s)

Manuscript Number: RSOS-181664

Title: Fabrication and characterization of capric acid/reduced graphene oxide decorated diatomite composite phase change materials for solar energy storage

The authors fabricated a novel type of CPCMs with CA and rGO decorated diatomite using vacuum impregnation. The structure, morphology and thermal properties of CPCMs had been studied in depth. The characterization and explanation made sense. I recommend publication in Royal Society Open Science after minor revisions. Some comments are provided as follows:

(1) Configuration and arrangement of figures are poor. There are so many figures in the draft manuscript. It is suggested some of the figures (e.g. Figure 6 (a) and (b), Figure 9 and 10) be combined.

(2) More details of the diatomite sample are required.

(3) The determination of the transition temperatures from the DSC thermograms should be specified (were they determined at the peak maximum?).

Review form: Reviewer 2

Is the manuscript scientifically sound in its present form?

Yes

Are the interpretations and conclusions justified by the results?

Yes

Is the language acceptable?

No

Is it clear how to access all supporting data?

Yes

Do you have any ethical concerns with this paper?

No

Have you any concerns about statistical analyses in this paper?

No

Recommendation?

Major revision is needed (please make suggestions in comments)

Comments to the Author(s)

In this study, DI-based CPCMs with good shape stability and high thermal conductivity were well designed. The microstructures, chemical compatibility, thermal conductivity,

thermophysical properties and shape-stability and were discussed in detail. This is a comprehensive and informative research and has certain significance in improving the properties of PCMs. However, there are some problems in the manuscript that should be further explained for publication in the present form. The following points need to be noticed:

1. The language must be improved, and there are too many shorts for materials throughout the full text made it difficult to understand.
2. The article emphasizes the DI-based CPCMs aims in solar energy storage but is not reflected in the manuscript. How can this material used in solar energy storage? And there also should make a comparison between this new DI-based CPCMs and the other current materials.
3. The authors studied shape-stability of the composite PCMs by leakage tests. What was the package capacity of composite PCMs without obviously liquid leakage? For the discussion in 4.2, what's the reason for the conclusion expressed as "Thus, the impregnation ratio of DI was improved through alkali treatment, and CA/rGO-DI (60wt. %: 40wt. %) was considered the optimum combination for producing CPCM."?
4. The reasons why CA/rGO-DI CPCMs exhibited the highest thermal conductivity should be further analyzed and clarified.
5. A more comprehensive and clearly conclusion should be supplemented.

Review form: Reviewer 3 (Yangai Liu)

Is the manuscript scientifically sound in its present form?

Yes

Are the interpretations and conclusions justified by the results?

Yes

Is the language acceptable?

Yes

Is it clear how to access all supporting data?

Yes

Do you have any ethical concerns with this paper?

No

Have you any concerns about statistical analyses in this paper?

No

Recommendation?

Accept with minor revision (please list in comments)

Comments to the Author(s)

This paper mainly reports that diatomite-based composite phase change materials (DI-based CPCMs) were fabricated by the vacuum impregnation of capric acid (CA) into reduced graphene oxide decorated diatomite (rGO-DI). Results shows that the CPCMs of this manuscript have the potential to be used in solar energy storage systems. On the whole, the experiments in the manuscript support the arguments well and this paper is reasonable in structure and expression basically. The photographs are all reasonably processed and exhibit legible information. In my opinion, this manuscript can be accepted for publication after revising.

1. In Fig. 2(c), it would be more convenient if you put 170°C to 250°C intervals with two vertical

dashed lines.

2. In Fig. 3, Some of the pictures have shadows, so you can make them more clearly. The luminescence properties of phosphors mentioned in your title is not complete, please add more.
3. In Fig.5, please add the legend.
4. In Fig.6, If there is a standard card of a single phase of CA, please put it in the picture for comparison.

Decision letter (RSOS-181664.R0)

19-Nov-2018

Dear Dr Li:

Title: Fabrication and characterization of capric acid/ reduced graphene oxide decorated diatomite composite phase change materials for solar energy storage
Manuscript ID: RSOS-181664

The editor assigned to your manuscript has now received comments from reviewers. We would like you to revise your paper in accordance with the referee and Subject Editor suggestions which can be found below (not including confidential reports to the Editor). Please note this decision does not guarantee eventual acceptance.

Please submit your revised paper before 12-Dec-2018. Please note that the revision deadline will expire at 00.00am on this date. If we do not hear from you within this time then it will be assumed that the paper has been withdrawn. In exceptional circumstances, extensions may be possible if agreed with the Editorial Office in advance. We do not allow multiple rounds of revision so we urge you to make every effort to fully address all of the comments at this stage. If deemed necessary by the Editors, your manuscript will be sent back to one or more of the original reviewers for assessment. If the original reviewers are not available we may invite new reviewers.

Please also include the following statements alongside the other end statements. As we cannot publish your manuscript without these end statements included, if you feel that a given heading is not relevant to your paper, please nevertheless include the heading and explicitly state that it is not relevant to your work.

• Ethics statement

Please clarify whether you received ethical approval from a local ethics committee to carry out your study. If so please include details of this, including the name of the committee that gave consent in a Research Ethics section after your main text. Please also clarify whether you received informed consent for the participants to participate in the study and state this in your Research Ethics section.

OR

Please clarify whether you obtained the necessary licences and approvals from your institutional animal ethics committee before conducting your research. Please provide details of these licences and approvals in an Animal Ethics section after your main text.

OR

Please clarify whether you obtained the appropriate permissions and licences to conduct the fieldwork detailed in your study. Please provide details of these in your methods section.

RSC Associate Editor:
Comments to the Author:
(There are no comments.)

RSC Subject Editor:
Comments to the Author:
(There are no comments.)

Reviewers' Comments to Author:
Reviewer: 1

Comments to the Author(s)

Manuscript Number: RSOS-181664

Title: Fabrication and characterization of capric acid/reduced graphene oxide decorated diatomite composite phase change materials for solar energy storage

The authors fabricated a novel type of CPCMs with CA and rGO decorated diatomite using vacuum impregnation. The structure, morphology and thermal properties of CPCMs had been

studied in depth. The characterization and explanation made sense. I recommend publication in Royal Society Open Science after minor revisions. Some comments are provided as follows:

- (1) Configuration and arrangement of figures are poor. There are so many figures in the draft manuscript. It is suggested some of the figures (e.g. Figure 6 (a) and (b), Figure 9 and 10) be combined.
- (2) More details of the diatomite sample are required.
- (3) The determination of the transition temperatures from the DSC thermograms should be specified (were they determined at the peak maximum ?).

Reviewer: 2

Comments to the Author(s)

In this study, DI-based CPCMs with good shape stability and high thermal conductivity were well designed. The microstructures, chemical compatibility, thermal conductivity, thermophysical properties and shape-stability and were discussed in detail. This is a comprehensive and informative research and has certain significance in improving the properties of PCMs. However, there are some problems in the manuscript that should be further explained for publication in the present form. The following points need to be noticed:

1. The language must be improved, and there are too many shorts for materials throughout the full text made it difficult to understand.
2. The article emphasizes the DI-based CPCMs aims in solar energy storage but is not reflected in the manuscript. How can this material used in solar energy storage? And there also should make a comparison between this new DI-based CPCMs and the other current materials.
3. The authors studied shape-stability of the composite PCMs by leakage tests. What was the package capacity of composite PCMs without obviously liquid leakage? For the discussion in 4.2, what's the reason for the conclusion expressed as "Thus, the impregnation ratio of DI was improved through alkali treatment, and CA/rGO-DI (60wt. %: 40wt. %) was considered the optimum combination for producing CPCM."?
4. The reasons why CA/rGO-DI CPCMs exhibited the highest thermal conductivity should be further analyzed and clarified.
5. A more comprehensive and clearly conclusion should be supplemented.

Reviewer: 3

Comments to the Author(s)

This paper mainly reports that diatomite-based composite phase change materials (DI-based CPCMs) were fabricated by the vacuum impregnation of capric acid (CA) into reduced graphene oxide decorated diatomite (rGO-DI). Results shows that the CPCMs of this manuscript have the potential to be used in solar energy storage systems. On the whole, the experiments in the manuscript support the arguments well and this paper is reasonable in structure and expression basically. The photographs are all reasonably processed and exhibit legible information. In my opinion, this manuscript can be accepted for publication after revising.

1. In Fig. 2(c), it would be more convenient if you put 170°C to 250°C intervals with two vertical dashed lines.
2. In Fig. 3, Some of the pictures have shadows, so you can make them more clearly. The luminescence properties of phosphors mentioned in your title is not complete, please add more.
3. In Fig.5, please add the legend.
4. In Fig.6, If there is a standard card of a single phase of CA, please put it in the picture for comparison.

Author's Response to Decision Letter for (RSOS-181664.R0)

See Appendix A.

Decision letter (RSOS-181664.R1)

03-Dec-2018

Dear Dr Li:

Title: Fabrication and characterization of capric acid/ reduced graphene oxide decorated diatomite composite phase change materials for solar energy storage
Manuscript ID: RSOS-181664.R1

It is a pleasure to accept your manuscript in its current form for publication in Royal Society Open Science. The chemistry content of Royal Society Open Science is published in collaboration with the Royal Society of Chemistry.

RSC Associate Editor
Comments to the Author:
(There are no comments.)

Reviewer(s)' Comments to Author:

Appendix A

List of Responses

Dear Editor:

Thank you for the opportunity to resubmit our revised manuscript entitled “Fabrication and characterization of capric acid/ reduced graphene oxide decorated diatomite composite phase change materials for solar energy” (ID: RSOS-181664). We have studied comments from you and reviewers carefully and have made correction. Those comments are all valuable and very helpful for revising and improving our paper, as well as the important guiding significance to our researches. The main corrections in the paper and the responds to the reviewer’s comments are as flowing:

Reviewer #1:

Response to comment 1: Configuration and arrangement of figures are poor. There are so many figures in the draft manuscript. It is suggested some of the figures (e.g. Figure 6 (a) and (b), Figure 9 and 10) be combined.

Response: Considering the reviewer’s suggestion, several figures in this paper has been redrawn (Fig. 2(c), Fig. 3, Fig. 5, Fig. 6) and combined

(Fig. 9 and 10). The intensity of CA is too high, if combine Fig. 6 (a) and (b), the Fig.6 will turned the following picture.

Response to comment 2: More details of the diatomite sample are required.

Response: We are sorry for our negligence. We added the details of the diatomite sample in Section 3.1, Table 1

Table 1 Chemical compositions (wt.%) of diatomite used in this study

Constituent	SiO ₂	Al ₂ O ₃	Fe ₂ O ₃	MgO	CaO	Na ₂ O	K ₂ O	TiO ₂	Other
Ratio (%)	80.21	10.26	2.18	0.88	0.9	2.22	0.8	0.3	2.1

Response to comment 3: The determination of the transition temperatures from the DSC thermograms should be specified (were they determined at the peak maximum?).

Response: Generally, the phase change temperature of PCMs were the onset temperature in DSC curves as the below shown, the peak

temperature was corresponding the endothermic/ exothermic peak temperature.

DSC curves of pure CA.

Special thanks to you for your good comments

Reviewer #2:

Response to comment 1: The language must be improved, and there are too many shorts for materials throughout the full text made it difficult to understand.

Response: The language of our manuscript has been refined and polished by a professional editing company.

Response to comment 2: The article emphasizes the DI-based CPCMs aims in solar energy storage but is not reflected in the manuscript. How can this material used in solar energy storage? And there also should make a comparison between this new DI-based CPCMs and the other

current materials.

Response: Considering the reviewer's suggestion, the illustration of PCM used in solar energy storage areas has been added in "Section 2 Introduction".

We had made a comparison between the DI-based CPCMs and other DI-based CPCMs in previous manuscript (Table 3) and considering the reviewer's suggestion, we made a comparison between the DI-based CPCMs and other clay-based CPCMs in the revision manuscript (Table 4). In previous research, the phase change latent of clay-based composite PCM is an important factor in solar energy storage. The excellent thermal properties of CA/rGO-DI FSPCMs could potentially be used as solar energy storage material.

Response to comment 3: The authors studied shape-stability of the composite PCMs by leakage tests. What was the package capacity of composite PCMs without obviously liquid leakage? For the discussion in 4.2, what's the reason for the conclusion expressed as "Thus, the impregnation ratio of DI was improved through alkali treatment, and CA/rGO-DI (60wt.%: 40wt.%) was considered the optimum combination for producing CPCM."?

Response: In this study, the highest content of PCM was 60wt.% with no leakage happening, in Fig.3 we can find that the composite PCMs were agglomerate when the content of CA was higher than 60wt.%, the composite PCMs were leaked after heating under CA content of 65wt.%. In order to indicate the package capacity of composite PCMs without obviously liquid leakage more obviously, the Fig.3 was redrawn, the mass loss of composite PCMs during the melting process was tested and the discussion in 4.2 was rewritten. The results as following:

4.2 Exudation stability of CA/DI and CA/rGO-DI CPCM

The macroscopic photographs of CA/DI and CA/rGO-DI-2 CPCM with different content of CA were shown in Fig.3(a), it is obvious that the composite PCMs were agglomerate when the content of CA was 65wt.%. In order to investigate the exudation stability, CA/DI and CA/rGO-DI-1 CPCM the were heated to 80°C for 0-2h, the residual quantity of CA/DI and CA/rGO-DI-1 CPCM were shown in Fig.3(b). The mass loss during the heating process can be neglected when the content of CA was 55wt.% and 60wt.%. Yet significant mass loss (about 10wt.%) occurred for CA/DI and CA/rGO-DI-1 CPCM when the content of CA was 65wt.%. Thus, the optimum mass fraction of CA in

composite without leakage is 60wt.%.

Fig. 3 Photographs of CA/DI and CA/rGO-DI-1 CPCM (a); The residual quantity of the composite PCMs during the heating process(b).

Response to comment 4: The reasons why CA/rGO-DI CPCM exhibited the highest thermal conductivity should be further analyzed and clarified.

Response: Considering the reviewer's suggestion, we further analyzed the reasons why CA/rGO-DI CPCM exhibited the highest thermal

conductivity.

Response to comment 5: A more comprehensive and clearly conclusion should be supplemented.

Response: We had rewritten the conclusion.

Special thanks to you for your good comments

Reviewer #3:

Response to comment 1: In Fig. 2(c), it would be more convenient if you put 170°C to 250°C intervals with two vertical dashed lines.

Response: Considering the reviewer's suggestion, two vertical dashed lines at 170°C and 250°C had been added in Fig. 2(c)

Response to comment 2: In Fig. 3 , Some of the pictures have shadows, so you can make them more clearly. The luminescence properties of phosphors mentioned in your title is not complete, please add more.

Response: Considering the reviewer's suggestion, we had redrawn the Fig. 3 and rewritten the "Section 4.2". In addition, the illustration of PCM used in solar energy storage areas has been added in "Section 2 Introduction"

Response to comment 3: In Fig.5, please add the legend.

Response: We are sorry for our negligence, the legend has been added

in Fig.5

Response to comment 4: In Fig.6, If there is a standard card of a single phase of CA, please put it in the picture for comparison.

Response: The Fig.6 has been rewritten, the diffraction peaks of pristine CA and DI has been showed in Fig.6. However, we had not found the standard card of a single phase of CA in previous paper and standard PDF card. In this paper, compared the XRD curves of CA, DI and CA/DI, the diffraction peaks of CA and DI were observed in CA/DI, which suggested that CA was successfully loaded into the supporting materials (DI or rGO-DI).

Special thanks to you for your good comments

We tried our best to improve the manuscript and made some changes in the manuscript. These changes will not influence the content and framework of the paper.

We hope that the revised manuscript is now acceptable for publication.

Once again, thank you very much for your comments and suggestions.